# SAT: Dynamic Spatial Aptitude Training for Multimodal Language Models

**Arijit Ray,**[1] **Jiafei Duan,**[2†] **Ellis Brown,**[5†] **Reuben Tan,**[1,4] **Dina Bashkirova,**[1]
**Rose Hendrix**,[3] **Kiana Ehsani,**[3] **Aniruddha Kembhavi,**[2,3] **Bryan A. Plummer,**[1]
**Ranjay Krishna,**[2,3*] **Kuo-Hao Zeng,**[3*] **Kate Saenko**[1*]

[1]Boston University,  [2]University of Washington,
[3]Allen Institute for AI,  [4]Microsoft Research,  [5]New York University

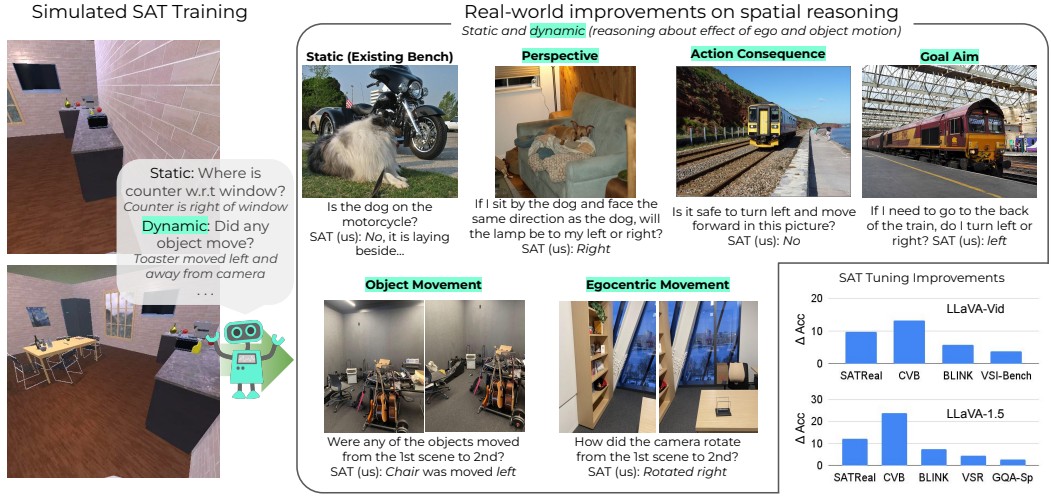

Figure 1: We propose Spatial Aptitude Training (SAT), which generates simulated spatial training data that allows us to go beyond simple static relationships in existing datasets. Inspired by cognitive science, we add more challenging **dynamic** questions that require reasoning about ego and object motion. SAT improves performance on both existing spatial benchmarks and dynamic reasoning on our new real-image benchmark.

## Abstract

Reasoning about motion and space is a fundamental cognitive capability that is required by multiple real-world applications. While many studies highlight that large multimodal language models (MLMs) struggle to reason about space, they only focus on *static* spatial relationships, and not *dynamic* awareness of motion and space—*i.e.,* reasoning about the effect of egocentric and object motions on spatial relationships. Manually annotating such object and camera movements is expensive. Hence, we introduce SAT, a simulated spatial aptitude training dataset comprising both static and dynamic spatial reasoning across 175K question-answer (QA) pairs and 20K scenes. Complementing this, we also construct a small (150 image-QAs) yet challenging dynamic spatial test set using real-world images. Leveraging our SAT datasets and 6 existing static spatial benchmarks, we systematically investigate what improves both static and dynamic spatial awareness. Our results reveal that simulations are surprisingly effective at imparting spatial aptitude to MLMs that translate to *real* images. We show that perfect annotations in simulation are more effective than existing approaches of pseudo-annotating real images. For instance, SAT training improves a LLaVA-13B model by an average 11% and a LLaVA-Video-7B model by an average 8% on multiple spatial benchmarks, including our real-image

*equal advising, †joint second author

dynamic test set and spatial reasoning on long videos—even outperforming some large proprietary models. While reasoning over static relationships improves with synthetic training data, there is still considerable room for improvement for dynamic reasoning questions.

# 1 Introduction

Reasoning about space and motion is a fundamental cognitive capability that allows humans and animals to survive and operate in the real world (Gardner, 2011; Vasilyeva & Lourenco, 2012; Blades & Spencer, 1994). While simple visual relationships (such as left-right) may be trivial to recognize for humans, more dynamic reasoning involving motion such as taking perspectives and reasoning about the effect of ego-motion or object movement requires more cognitive support (Gardner, 2011; Tversky, 2019). Hence, it is not surprising that such "dynamic" spatial reasoning capabilities in children correlate with their aptitude in geometry, physics, and even linguistic reasoning (Taylor & Tversky, 1992; Mallot, 2024).

Despite the promise of multimodal language models (MLMs) as general-purpose intelligent agents (Achiam et al., 2023; Zhu et al., 2023; Liu et al., 2024), recent studies reveal that these models still struggle with spatial reasoning (Chen et al., 2024; Cheng et al., 2024; Tong et al., 2024; Fu et al., 2024b). Specifically, MLMs often fail to predict the relative positions of objects in static images—a limitation attributed to the scarcity of spatial relationship annotations in their training data. To address this, recent methods (Cheng et al., 2024; Chen et al., 2024) introduce pseudo-annotations to encode static object relationships in real images (Cheng et al., 2024), which require extensive engineering and refining to scale. Furthermore, many real-world applications require reasoning that extends beyond static object positions. For example, smart glasses and embodied AI applications need to reason about *dynamic* scenes, where reasoning about the effect of object and camera movements are essential (Duan et al., 2024; Yuan et al., 2024a; Liu et al., 2025). Generating pseudo-annotations using existing methods for such dynamic settings is non-trivial due to the complexity of predicting action causality on real images.

We explore the possibility of using simulation data as a simple solution to impart both static and dynamic spatial reasoning into MLMs. We propose **Spatial Aptitude Training** (SAT), an approach to generate spatial question-answer (QA) data without any human supervision. Manually annotating 3D movements is expensive; hence, SAT leverages 22K ProcTHOR (Deitke et al., 2022) scenes composed of 1K assets to generate 175K QA pairs. With perfect 3D information and control of the assets, SAT goes beyond static object relationships to questions that require reasoning about egocentric and object movements. Since our data is generated procedurally by composing assets, it can be scaled up arbitrarily. Hence, SAT is programmatically controllable, unlike existing 3D (Brazil et al., 2023; Azuma et al., 2022) or spatial datasets (Cheng et al., 2024), which are also not compositional.

With SAT, we analyze what types of training data improve spatial reasoning. We focus on two types of spatial reasoning data. First, spatial-QA about object relations in static scenes, shown in Fig. 1 (under "Existing Spatial Bench"), to impart reasoning about the relative locations of objects in the scene (*e.g.,* where is object X with respect to object Y?). Next, we evaluate the effect of dynamic spatial tasks, Fig. 1 (shown under "Our SAT Dynamic Bench"). This includes questions about egocentric movement, object movement, allocentric perspective, goal aiming, and action consequences. These dynamic spatial reasoning skills are well-studied in human cognitive development: children understand the consequences of self-motion (*e.g.,* the moving room test (Anderson et al., 2013)), track how entities move in a scene (Anderson et al., 2013), and can adopt others' viewpoints when planning actions (Brucato et al., 2023; Vasilyeva & Lourenco, 2012; Blades & Spencer, 1994).

We use two widely adopted open-source MLMs, an image-based LLaVA-1.5-13B and a video-based LLaVA-Video-7B, as our base models for evaluations. To test static spatial reasoning, we use four contemporary real image benchmarks: CV-Bench (Tong et al., 2024), BLINK (Fu et al., 2024b), Visual Spatial Relations (VSR) dataset (Liu et al., 2023), and GQA-Spatial (Hudson & Manning, 2019a). In addition, we test the effect of SAT on spatial

Table 1: Comparison of existing datasets to ours. Our training dataset is synthetic and interactive, allowing us to generate spatial QA and extend to dynamic (*e.g.,* object movement) reasoning data for free. It can also be easily extended to generate new scenes and tasks.

| | SAT (Ours) | 2D Vision-Language GQA, VG, Obj365 | 3D Vision-Langauge Omni3D, ScanQA | Spatial Rel VSR, 2.5VRD | Spatial QA CVBench, BLINK, Sp VLM, Sp RGPT |
|---|---|---|---|---|---|
| **2D Annotations** | ✓ | ✓ | ✓ | ✓ | ✓ |
| **3D Annotations** | ✓ | ✗ | ✓ | ✓(2.5D) | ✗ |
| **Static Spatial QA** | ✓ | ✓ | ✓ | ✗ | ✓ |
| **Dynamic Spatial QA** | ✓ | ✗ | ✗ | ✗ | ✗ |
| **New Scene/Task Gen** | ✓ | ✗ | ✗ | ✗ | ✗ |
| **Object Interaction** | ✓ | ✗ | ✗ | ✗ | ✗ |

reasoning on longer videos using the recent VSI-Bench (Yang et al., 2024a). Since no dataset exists for dynamic spatial reasoning, we construct a difficult SAT test set on real images.

Challenging conventional wisdom about sim-to-real transfer in model training, we find simulations to be surprisingly effective at imparting spatial intelligence to MLMs. Our simulated SAT training improves the baseline LLaVA-13B model by 11% and LLaVA-video-7B by 8% (absolute) on average on a wide range of spatial benchmarks, including our challenging dynamic test set—even outperforming some larger closed-source and spatially-tuned models (Achiam et al., 2023; Team et al., 2023). Interestingly, SAT also improves spatial reasoning on long videos (Yang et al., 2024a)—especially in route planning by 4%, showing promise for aiding embodied applications. However, while simple static relationships in existing datasets are easy to improve, our best model performs at 62% on our SAT dynamic questions (random chance is 50%)—hence, leaving considerable room for improvement.

Our work shows promise that training with data generated using embodied movements and interactions in simulators can indeed help instill spatial intelligence in MLMs.

## 2 Related work

Our work draws inspiration from fundamental schools of thought in neuroscience that suggest spatial intelligence is a core foundation for most cognitive abilities (Tversky & Suwa, 2009; Mallot, 2024; Duan et al., 2022b).

**Spatial understanding benchmarks.** We visualize a comparison of our dataset with existing 3D and spatial benchmarks (Table 1). 3D understanding is vital for various computer vision tasks, including segmentation (Jatavallabhula et al., 2023; Hong et al., 2023a; Kerbl et al., 2023; Nie et al., 2020; Kundu et al., 2022; Wang, 2023), object localization (Qi et al., 2019; Sajjadi et al., 2022; Kamath et al., 2021), tracking (Agia et al., 2022; Kurenkov et al., 2023; Li et al., 2022), fine-grained scene captioning (Chen et al., 2021b; Su et al., 2021), open-vocabulary classification (Chen et al., 2020; Hong et al., 2023a; Achlioptas et al., 2020; Shao et al., 2019), and question answering (Ye et al., 2021; Azuma et al., 2022; Linghu et al., 2024). Most use 3D scans of real-world data (Hong et al., 2023b), but 3d scans are expensive to recompute for dynamic scenes and most MLMs are trained to input 2D images-hence we focus on 3D spatial reasoning on 2D inputs. In light of potential applications in embodied AI (Duan et al., 2022b), we align our dataset to be compatible with physics simulators, allowing us to go beyond datasets which only leverage image annotations (Caesar et al., 2020) and have static spatial reasoning (Fu et al., 2024b; Duan et al., 2022a; Chen et al., 2024; Kamath et al., 2023; Su et al., 2021; Liu et al., 2023; Goyal et al., 2020; Yang et al., 2019; Du et al., 2024).

**Synthetic training data.** Many have investigated whether synthetic information can boost reasoning in real environments Geng et al. (2024), finding that synthetic images sometimes help in classification (Chen et al., 2021a), semantic understanding (Mishra et al., 2022), correcting biases (Qraitem et al., 2023), and teaching navigation capabilities to embodied AI (Ehsani et al., 2024; Silwal et al., 2024). Inspired by this, our work explores if synthetic data with perfect 2D/3D information can improve spatial reasoning.

**Vision-language models.** Our task is heavily influenced by the emergence of multimodal foundation models (Radford et al., 2021; Jia et al., 2021; Wang et al., 2022a; Yuan et al., 2021;

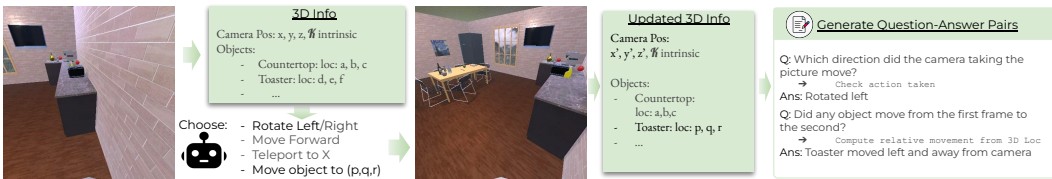

Figure 2: Method of generating our SAT dynamic data: we take actions in a 3D simulator and check the 3D locations of assets. We generate template-based QA pairs based on how the 3D nature of the scene changes with actions taken.

Fu et al., 2021; Wang et al., 2022b; Ye et al., 2023). VLMs have been adapted for a wide range of downstream image (Tsimpoukelli et al., 2021; Driess et al., 2023; Zhu et al., 2023; Li et al., 2022; Alayrac et al., 2022; Dai et al., 2024) and video (Zhang et al., 2023; Maaz et al., 2023; Tan et al., 2024; Wang et al., 2024; Li et al., 2023; Luo et al., 2023) understanding tasks. Many tasks require spatial understanding of the scene (Brohan et al., 2023; Zhang et al., 2024b). Multiple recent works (Fu et al., 2024a; Hong et al., 2023a; Cho et al., 2024) have noted their weakness in 3D pose and location estimation. Adjacent to works like Ray et al. (2023); Zhang et al. (2024a); Hsieh et al. (2023); Thrush et al. (2022) that point to deficiencies in understanding compositions of spatial relationships with objects and attributes, we explore if perfect 3D information in synthetic images can improve spatial understanding.

# 3 SAT: Spatial Aptitude Training

Our goal is to improve the spatial reasoning capabilities of MLMs. Since obtaining varied 3D annotations with interactive movements of the camera and objects on real images is expensive and tedious, we procedurally generate photo-realistic environments and automatically curate training data. The resulting data generation pipeline, SAT, serves as both instruction-tuning data for MLMs and as a benchmark to test the dynamic spatial reasoning capabilities not present in existing benchmarks.

In total, our training dataset contains 175K questions across 22K procedurally generated scenes from ProcTHOR-10K dataset of indoor apartment buildings. Next, we outline the generation process for the two kinds of spatial questions—static and dynamic.

## 3.1 Generating Static Spatial QAs.

Aligning with contemporary benchmarks for spatial reasoning, we first generate instruction-tuning data for static spatial reasoning that deal with relative relations of objects. Overall, we generate 127K static spatial QA pairs across 8K images across the following types:

**Relative spatial relations.** We generate questions about the relative location of one object in the scene to other objects. We form two kinds of questions—(1) judging if object X is to the left, right, above, or below object Y. For example, *Is the wine bottle to the left or right of the plate?* (2) judging if object A or B is closer to another object C. For example, *Which object is closer to the wine bottle—the cup, or the plate?* Given the camera parameters, we first project the objects' poses into the camera coordinate system and then generate the corresponding answers. More details about the camera coordinate normalization are in the appendix.

**Relative Depth.** We generate questions about judging whether object X is closer to the camera than object Y by calculating the distances to the objects in the simulator. For example, *Is the wine bottle closer to the camera than the plate?*

**Count.** Since many MLMs (Goyal et al., 2017) struggle with counting, we include counting questions using the number of object instances from the metadata of our simulator.

### 3.2 Generating Dynamic Spatial QA

Grounded in spatial cognitive tests (Anderson et al., 2013; Brucato et al., 2023; Vasilyeva & Lourenco, 2012), we outline five different complex tasks that require reasoning about egocentric movements, object movements, and allocentric perspectives. We generate 48K QAs on 13K images. The high-level idea behind generating such QAs is illustrated in Figure 2. Given a frame in a simulated environment, we take an action and formulate QAs based on how the 3D orientation of objects changes based on the action taken. Below, we outline the specific approach for each of the question types.

**Egocentric Movement.** This is based on the "moving room test" Anderson et al. (2013), a fundamental test designed to assess and improve spatial cognitive development in children. This test aims to measure if an agent can judge how they moved given two frames. This is useful beyond just measuring spatial cognition since this task can help judge navigation trajectories from egocentric videos in embodied applications. We take a random action from the choices of rotating left or right by a random angle, or moving left/right by a random distance. Note that *moving* and *rotating* are different since a camera can be moving left while rotating right. Based on the action taken, we formulate a question of the type: *How did the camera taking the video likely rotate/move?* with the answer being the action sequence taken. We have 6.9K training image-QA pairs of this type. This is denoted as **EgoM** in the tables. More details of the exact simulator actions taken to generate this are in the appendix.

**Object Movement.** Here, we randomly choose an object and move it by a randomly chosen distance and direction, ensuring that the object is still in the frame of view. Next, we compare the updated 3D position of the object with the original position normalized by the camera coordinates to decide if the object got closer, further, more left, or more right from the original position, or if it did not move. Based on that, we form QA pairs of the type: *Did any of the objects move from the first frame to the second frame?* with the answer being the way the object moved. Note that sometimes objects may move in conjunction with camera movement. To answer the questions accurately, the agent needs to learn to distinguish between egocentric movement and objects moving. We have 6.9K training image-QA pairs of this type. This is denoted as **ObjM** in the tables.

**Allocentric Perspective.** Inspired by a spatial cognitive test for humans and animals Brucato et al. (2023), this test checks if the agent is able to take the perspective of another viewer and judge the relative locations of objects according to the other viewer. To make such reasoning QAs, we first choose a 2D point in the scene and mark it as "X". Next, we teleport that agent to the 3D location corresponding to "X" (determined by ray tracing). We check the relative positions of objects according to the camera view from "X" (similarly as described in 3.1. We make QA's of the type: *For someone at the mark 'X' facing left/right by 90 degrees, would the <object> be to their left or right?* We have 6K training image-QA pairs of this type. This is denoted as **Pers** in the tables.

**Goal Aiming.** Aiming is a prerequisite for efficient navigation to objects Franz & Mallot (2000), a fundamental spatial cognitive capability. Hence, we design QAs that check how well agents can aim to the desired object. We pick a random object and calculate the angle of the object to the camera using the 3D location of the object and camera, assuming looking forward is 0 degrees (exact equations in the supplementary). Based on the angle, we formulate questions of the type: *I need to go to the countertop. Which direction should I turn to face it?* Since precise angles are hard to judge from a single image, we formulate answers as choices of rough angles to turn towards the left or right. We have 6.8K training image-QA pairs of this type. This is denoted as **Aim** in the tables.

**Action Consequence.** Here, the agent needs to reason about how the spatial relationships change when it takes a movement action, inspired by how humans can reason about the consequences of their movements in an environment Franz & Mallot (2000). Here, we show the first frame and ask the agent to judge if we would move closer/further, or look towards or away from an object if it took that action. *e.g., If I rotate left by 90 degrees and move forward, would I move further from the sofa?* Note that in most cases, moving forward would get us closer to an object. To make the distribution of answers even, we sometimes rephrase the question as to whether we would be facing the object or not. We have 15K training image-QA pairs of this type. This is denoted as **EgoAct** in the tables.

Table 2: Both large proprietary and open-source MLMs struggle on our dynamic SAT spatial test set, including strong models like Gemini1.5-pro and spatially-tuned Robopoint. SAT training improves LLaVA models significantly on SAT test set.

| | SAT Real | SAT Synthetic | | | | | |
|---|---|---|---|---|---|---|---|
| | Avg | EgoM | ObjM | EgoAct | GoalAim | Pers | Avg |
| a. GPT4-V | 50.7 | 54.7 | 32.7 | 52.0 | 50.5 | 34.2 | 44.8 |
| b. GPT4-o | 57.5 | 61.5 | 33.2 | 47.6 | 67.5 | 37.5 | 49.4 |
| c. Gemini1.5-flash | 57.6 | 67.1 | 33.1 | 52.9 | 64.0 | 32.7 | 50.0 |
| d. Gemini1.5-pro | **64.8** | 57.7 | 29.8 | 55.5 | 56.9 | 49.5 | 49.9 |
| e. Robopoint-13B | 46.6 | 50.2 | 69.4 | 48.8 | 72.6 | 25.5 | 53.3 |
| f. LLaVA-1.5-13B | 41.6 | 46.6 | 73.8 | 49.7 | 45.6 | 39.9 | 51.1 |
| g.   + SAT | 54.9 | 61.7 | **90.2** | **91.4** | **96.8** | **98.5** | **87.7** |
| Δ *Improvement* | +13.3 | +15.1 | +16.4 | +41.7 | +51.2 | +58.6 | +36.6 |
| h. LLaVA-Video-7B | 53.5 | 56.4 | 82.7 | 48.0 | 52.9 | 47.1 | 57.4 |
| i.   + SAT | 63.4 | **79.6** | 80.4 | 85.3 | 56.4 | 88.4 | 78.0 |
| Δ *Improvement* | +9.9 | +23.2 | -2.3 | +37.4 | +3.5 | +41.3 | +20.6 |

**Real-image Test Set.** Existing image spatial datasets only test for static relationships and not the above-mentioned dynamic capabilities. A recent video spatial dataset (Yang et al., 2024a) also focuses on static object relations in a video input where the scene doesn't change. Hence, we construct a small yet challenging test set of dynamic QAs that focus on how relationships change given ego or object motions and varying perspectives. We use expert annotators to annotate 150 image-question-answer pairs ($\sim 30$ each of the above dynamic types). We also generate a larger synthetic test set—647 object movement, 647 egocentric movement, 592 goal aim, 1336 action consequence, and 778 perspective questions on 805 images for analysis on individual splits.

### 3.3   Finetuning MLMs with SAT

We provide the MLM with the required image(s) and the question and ask it to choose between two answers—a correct answer and a distractor answer, following existing benchmark standards (Fu et al., 2024b; Cai et al., 2024). The distractor answer is generated by switching the correct answer to the opposite/distractor word (*e.g.*, left → right, or "did not move"). To prevent catastrophic forgetting, we mix in some of the original pre-training data during tuning from the LLaVA Instruct-tune dataset (Liu et al., 2024). More tuning and prompt details are in the appendix.

## 4   Experiments

Using our SAT data as a training set and existing spatial benchmarks along with our SAT real-image dynamic test set, we investigate what encourages spatial aptitude in MLMs.

**Setup.** We use two popular open-source MLMs: LLaVA-1.5-13B (Liu et al., 2024) and LLaVA-Video-7B (Zhang et al., 2024d) for our experiments. We report the standard accuracy metric used for question-answering evaluations by checking if the predicted answer matches the GT answer. We notice better performance for off-the-shelf MLMs when we provide the options in text (*e.g.*, Choose between *right* or *left*) as opposed to option numbers (*e.g.*, Choose between option *A* or *B*). Hence, we sometimes report higher performances for the baselines than those reported by the original papers. Below, we present the key experimental questions we wish to ask.

### 4.1   Can SAT **improve spatial reasoning in MLMs?**

To evaluate the effect of SAT training on spatial performance, we use 4 existing static spatial benchmarks—CVBench (Tong et al., 2024), BLINK (spatial subsets) (Fu et al., 2024b), Visual

Table 3: SAT tuning improves performance of open MLMs on existing static spatial relationship benchmarks—often outperforming large proprietary and spatially-tuned Robopoint.

| | CVBench (Tong et al., 2024) | | | | | BLINK (Fu et al., 2024b) | | | |
|---|---|---|---|---|---|---|---|---|---|
| | Count | 2DRel | 3DDep | 3DDis | Avg | MV | RelDep | SpRel | Avg |
| a. GPT4-V | 62.4 | 71.1 | 79.8 | 68.3 | 70.2 | 55.6 | 59.7 | 72.7 | 62.7 |
| b. GPT4-o | 65.9 | 85.7 | 87.8 | 78.2 | **78.9** | **58.6** | 69.4 | **82.5** | **70.2** |
| c. Gemini1.5-flash | 66.0 | 76.9 | 75.3 | 68.3 | 71.4 | 51.1 | 62.9 | 62.9 | 59.0 |
| d. Gemini1.5-pro | **70.4** | 85.2 | 82.4 | 72.8 | 77.4 | 36.8 | 70.2 | 70.6 | 59.2 |
| e. Robopoint-13B | 53.6 | 79.4 | 74.7 | 71.3 | 69.1 | 48.1 | 51.6 | 75.5 | 58.4 |
| f. LLaVA-1.5-13B | 58.2 | 46.6 | 53.0 | 47.8 | 51.7 | 45.1 | 56.4 | 69.9 | 57.1 |
| g.  + SAT | 61.5 | **89.7** | 80.7 | 73.0 | 75.6 | 44.4 | **76.6** | 72.7 | 64.6 |
| Δ Improvement | +3.3 | +43.1 | +27.7 | +25.2 | +23.9 | -0.7 | +20.2 | +2.8 | +7.4 |
| h. LLaVA-Vid-7B | 59.3 | 77.0 | 71.3 | 54.7 | 65.2 | 39.1 | 55.6 | 75.5 | 56.7 |
| i.  + SAT | 66.2 | 81.2 | **88.2** | **79.3** | 78.4 | 48.1 | 66.1 | 73.4 | 62.6 |
| Δ Improvement | +6.9 | +4.2 | +16.9 | +24.6 | +13.2 | +9.0 | +10.5 | -2.1 | +5.8 |

Table 4: SAT also improves spatial performance on videos, VSI-Bench (Yang et al., 2024a)

| | Rel Dist | Rel Dir | Rt. Plan | App. Order | MC Avg |
|---|---|---|---|---|---|
| a. GPT4-o | 37.0 | 41.3 | 31.5 | 28.5 | 34.6 |
| b. Gemini-1.5-flash | 37.7 | 41.0 | 31.5 | 37.8 | 36.9 |
| c. Gemini-1.5-pro | **51.3** | **46.3** | 36.0 | 34.6 | **42.1** |
| d. LLaVA-Video-7B | 43.9 | 42.0 | 33.5 | 32.3 | 37.9 |
| e.  + SAT | 47.9 | 39.6 | **38.7** | **40.6** | 41.7 |
| Δ Improvement | +4.0 | -2.4 | +5.2 | +8.3 | +3.8 |

Spatial Relations (VSR) (Liu et al., 2023), and GQA-Spatial (Hudson & Manning, 2019a). For dynamic reasoning, we use our SAT test sets on both real and synthetic images. Since SAT real is small, we only report the overall average and report individual split performance on our synthetic test set for a more detailed analysis. Finally, we evaluate the effect of SAT tuning on a recent long-video spatial benchmark, VSI-Bench (Yang et al., 2024a), by tuning LLaVA-Video-7B (Zhang et al., 2024d).

**Open MLMs struggle on spatial reasoning. While proprietary MLMs perform better on static reasoning, they still struggle with dynamic reasoning.** In Tables 3 and 2, we see that open-source MLMs (rows f and h) struggle on both static relationships and dynamic reasoning. Larger closed-source models like GPT4-o (Achiam et al., 2023), Gemini-1.5-pro (Team et al., 2023) and spatially-tuned Robopoint (Yuan et al., 2024b) are better at static reasoning (in Table 3 row b, d), but still struggle on dynamic QAs (Table 2 rows a-e). On SAT synthetic, aiming at the goal is generally easier for spatially stronger models like RoboPoint and GPT4-o since it also mostly requires judging object position.

**Training on SAT significantly improves static spatial reasoning on real images for MLMs.** As shown in Table 3, tuning on SAT improves performance on static spatial questions for both LLaVA-1.5 and LLaVA-Video models by significant amounts (rows f vs g and h vs i). Comparing rows a-e, we observe that SAT tuning makes both LLaVA models match/outperform some closed-source models on zero-shot performance on CVBench and BLINK. Compared to an existing spatially-tuned baseline, we outperform RoboPoint-13B (Yuan et al., 2024b) (row e). This shows promise that SAT instruction-tuning may push performance further for these stronger models.

**Training on SAT improves dynamic spatial reasoning on real images.** Tuning on simulated SAT improves performance on SAT real test set as shown in Table 2 (column 1) by 12% for LLaVA-1.5 and by 10% for LLaVA-Video. This shows strong sim-to-real transfer. Once again, our strongest model matches Gemini-1.5-pro and outperforms GPT models.

**Camera movement and out-of-domain relations are hard to improve** BLINK SpRel has abstract relationships (especially with people) not present in our synthetic data (*e.g.,* "looking away", "surrounding"). Hence, in Table 3, gains are modest on SpRel (row f vs g), with

Table 5: SAT also improves out-of domain spatial relation (such as outdoor and video reasoning with humans on MME-RealWorld-Lite) despite having only indoor scenes.

| Model | Avg | Motion | Interaction | Position | Reasoning | Perception |
|---|---|---|---|---|---|---|
| LLaVA-Vid-7B | 35% | 26.8% | 24.5% | 34% | 36.8% | 35.0% |
| + SAT | 39% | 33.2% | 31.5% | 42% | 39.7% | 39.6% |
| Δ *Improvement* | *+4%* | *+6.4%* | *+7%* | *+8%* | *+2.9%* | *+4.6%* |

Table 6: Performance on some traditional spatial benchmarks- VSR (Liu et al., 2023), GQA-Sp (Hudson & Manning, 2019b)- and other VQA benchmarks- GQA (Hudson & Manning, 2019b), OK-VQA (Marino et al., 2019), VQAv2 (Goyal et al., 2017)- not focused on spatial relationships. Our spatial tuning remembers pre-training commonsense on other capabilities while improving spatial reasoning.

| | VSR | GQA-Sp | GQA | OK-VQA | VQAv2 |
|---|---|---|---|---|---|
| a. LLaVA-1.5-13B | 65.9 | 53.1 | **78.6** | 30.7 | 60.5 |
| b. + SAT | **70.4** | **55.8** | 71.8 | **35.1** | **60.9** |

LLaVA-Video (row h vs i) showing no improvement. Camera movements are also hard (BLINK MV). While more investigation is needed, we believe this may be due to translation invariance in ViT (Dosovitskiy et al., 2021) leading to minimal feature changes between images with subtle camera movement.

**Training on** SAT **improves spatial reasoning in videos.** Even though SAT only has up to 2 frames in its training data, we evaluate its impact on longer videos on Yang et al. (2024a). The results are shown in Table 4. We see that SAT training helps significantly on route planning capabilities (column Rt. Plan) while also improving other spatial splits. This shows promise that SAT tuning can benefit embodied AI applications.

SAT **training also improves on outdoor videos including humans.** SAT dataset is rendered using indoor simulators without any humans. However, when trained on SAT, we see improvements on outdoor video reasoning that includes humans on MME-Realworld-Lite (Zhang et al., 2024c) as shown in Table 5. While MME-RealWorld has numerous splits related to reasoning and perception, we highlight some of the splits where we see the most improvements when tuned on SAT along with the overall average (Avg column) and the average for all "Reasoning" and "Perception"-based splits.

**Our** SAT**-tuned model maintains its non-spatial reasoning capabilities.** We run an evaluation on some standard VQA benchmarks—namely, GQA (Hudson & Manning, 2019a), VQAv2 (Goyal et al., 2017), and OK-VQA (Marino et al., 2019) (9K image-QA pairs, random 3K from each). We maintain overall performance on them compared to the off-the-shelf LLaVA (Liu et al., 2024) as shown in Table 6. This suggests that we remember pre-trained vision-language commonsense while adding stronger spatial capabilities.

SAT **synthetic is a useful diagnostic test set** The lack of improvement on object relations (SpRel and 2DRel) in LLaVA-Video is also reflected by a lesser gain on the GoalAim (row h vs i, Table 2) on SAT synthetic test, where the model needs to judge object position. Similar for SAT EgoM (Table 2) and BLINK MV (Table 3) dealing with camera movements-LLaVA-Video improves more than LlaVA on both. Hence, despite being synthetic, SAT synthetic is also a useful diagnostic test set.

## 4.2 What are effective types of spatial training data?

**Sources of instruction tuning** Since contemporary work on spatial understanding uses pseudo annotations (Chen et al., 2024; Cheng et al., 2024), we wish to compare to such baselines as well. Since their datasets are not easily available yet, we reproduce a similar instruction-tuning set using pseudo annotations to create spatial QAs. We keep formats the same as SAT to ensure the performance differences are attributed to the source of spatial information and not the question formats.

Table 7: Table showing the effectiveness of SAT over existing sources of spatial data. Dynamic spatial reasoning data improves performance over just static spatial relationship data.

| | SATReal | CVBench | | | | | BLINK | | | | VSR |
|---|---|---|---|---|---|---|---|---|---|---|---|
| | | Count | 2D Rel | 3DDep | 3DDist | Avg | MV | RelDep | SpRel | Avg | |
| a. LLaVA-1.5 | 41.6 | 58.2 | 46.6 | 53.0 | 47.8 | 51.7 | 45.1 | 56.4 | 69.9 | 57.1 | 66.0 |
| b. + GQAPseudo | 45.1 | 63.1 | 64.0 | 41.2 | 48.7 | 54.5 | 53.4 | 48.4 | 57.3 | 53.1 | 62.3 |
| c. + VSR/25VRD | 45.4 | 60.8 | 64.2 | 54.3 | 44.7 | 55.9 | 3.0 | 61.3 | 69.9 | 44.7 | 67.9 |
| d. + SAT Static | 46.0 | 59.5 | 81.7 | 72.5 | 54.2 | 66.4 | **55.6** | 66.9 | 66.4 | 63.0 | 68.0 |
| e.   + Dynamic | **54.9** | **61.5** | **89.7** | **80.7** | **73.0** | **75.6** | 44.4 | **76.6** | **72.7** | **64.6** | **70.4** |

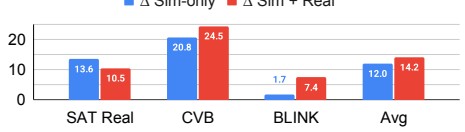

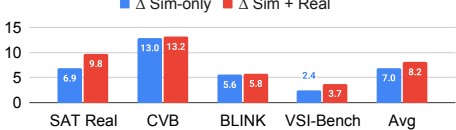

(a) Improvement over LLaVA-1.5-13B

(b) Improvement over LLaVA-Video-7B

Figure 3: Simulation-only training vs mixing in some original pre-training data. Sim-only is very effective, mixing in pre-training data improves further.

– GQAPseudo: We create SAT-Static like instruction-tuning data by inferring the depth (Yang et al., 2024b) on images from GQA (Hudson & Manning, 2019a) and VisualGenome (Krishna et al., 2017) and the 2D bounding box annotations. We filter out potential incorrect relationships using simple heuristics (more details in the appendix). We create 225K static spatial image-QA tuples. Generating pseudo-annotated dynamic QAs is not possible since we cannot easily infer the result of actions on real images, hence, we only generate static QAs.

– VSR/25VRD: We use spatial relationship datasets like VSR (Liu et al., 2023) and 2.5VRD (Su et al., 2021) to produce more real spatial QAs. These datasets contain relationships such as "touching", or "behind", allowing us to formulate QAs such as "Is the cat touching the sofa?" We combine VSR and 2.5VRD to create 200K image-QA tuples.

SAT **simulated data outperforms pseudo-annotations or human annotations.** In Table 7, we see that pseudo-annotations can improve performance on certain splits like Count and 2DRel on CVBench (row b). However, noise in annotations (*e.g.,* bounding boxes often not accounting for the entire object) leads to errors in judging depth of the entire object relative to other objects (*e.g.,* a part of the annotated "lake" may be in front of a "tree", but the majority of it may lie behind it). Following more complex careful curation similar to Cheng et al. (2024) may improve this performance, which we leave to future work since their data is not yet easily available. Human-annotated spatial relations (using Su et al. (2021); Liu et al. (2023)) tend to perform well in-domain on VSR test. While the higher diversity of human-annotated relations also helps prevent forgetting on BLINK SpRel (row c vs d), they are finite and not easily composable to generate varied 3D and dynamic data. Hence, we observe no improvements on SAT dynamic and minimal gains in 3D perception splits.

**Adding dynamic QAs further improves performance over just static QAs.** We notice improvements on both dynamic and static performance when we add dynamic data as noted in Table 7 (row d vs e). However, BLINK MV remains challenging and we observe no performance gain. While static-only seems to perform better on BLINK MV, we observe that simply predicting "moved right" all the time achieves that performance. We also notice gains if dynamic mixed data matches the amount of static-only (in appendix Table 10).

**SAT training scales with more data, especially when dynamic splits added.** Since SAT is generated procedurally, we wish to explore whether spatial performance of MLMs scale with more data. Hence, we vary the amounts of training data (train for 1 epoch). As shown in Figure 4, we see that while performance tends to saturate with static-only training (likely due to overfitting to simpler relationships), the performance scales positively with dynamic reasoning questions mixed in.

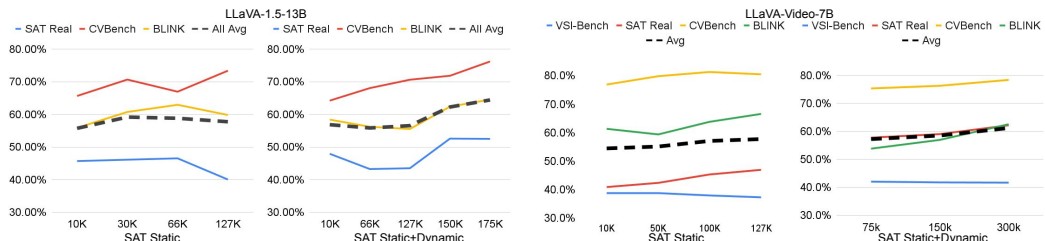

Figure 4: Ablations with training data size. We see that performance generally scales with data, with dynamic showing better scaling.

**Mixing in some original pretraining data when training in simulation helps performance**
We check whether simulation data alone suffices to impart spatial reasoning to MLMs. For *sim-only* training, we use our full SAT static+dynamic data. We compare this with *sim+real* training, where we mix in random samples of the original pretraining data 60% of the time during the training along with our SAT data. For both cases, the model sees the same amount of new SAT spatial data. As shown in Figure 3, surprisingly, simulation alone is still effective at improving the spatial awareness on real images. However, mixing in pretraining data improves further.

## 5 Discussion

**Limitations.** We instruction-tune two kinds of MLMs for spatial reasoning, an image-only LLaVA (Liu et al., 2024) and a video LLaVA-Video (Zhang et al., 2024d). While we remember pretraining commonsense as noted in Table 6, we haven't explored improving other capabilities like math and science reasoning Yue et al. (2024). The scope of the paper, however, is to analyze what kinds of data improve spatial performance, and not a large-scale training of a new MLM. Additionally, further analysis on more MLMs (Dubey et al., 2024; Deitke et al., 2024) might be beneficial.

**Future work.** Although our study focuses on evaluating the spatial reasoning capabilities of MLMs, it can be extended in various avenues. For instance, to determine the kinds of embodied applications that benefit from improved complex spatial reasoning. We do see gains in accuracy in route planning in VSI-Bench (Yang et al., 2024a), which should help embodied navigation. As a more direct evaluation, we also check the action prediction accuracy (from a choice of going left/right/forward) for a given frame on the SPOC Easy-ObjectNav benchmark (Ehsani et al., 2024). Our model (SAT Dynamic) scores an accuracy of 51% compared to 40% for training only on static spatial questions. This suggests that embodied navigation might benefit from improved dynamic spatial reasoning. We leave a more thorough evaluation in this direction as future work. Further, leveraging the interactive nature of our scenes with perfect 3D information could facilitate more explorations in dynamic and chain-of-thought causal reasoning. Our causal annotations in language and their corresponding image frames could also advance language-controlled world models Ball et al. (2025), a direction we leave for future work.

**Conclusion.** We propose a training and testing dataset of dynamic motion-based spatial tasks that go beyond simpler static perception of relationships in existing datasets. This improves MLMs on numerous spatial benchmarks while maintaining pre-trained commonsense. We hope that SAT paves the way for bridging the gap between passive perception and active interaction reasoning in MLMs, making them more suitable for deployment in real-life applications.

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

## A Appendix

**Acknowledgements**

This project was partially funded by Amazon Science and NSF award number 2120322. We wish to thank Devdeep Ray and Jang Hyun Cho for helpful discussions on 3D and depth estimation. The contents of the paper do not represent the views of the US Government or the funding agencies.

### A.1 Discussion

**Limitations.** We instruction-tune two kinds of MLMs for spatial reasoning, an image-only LLaVA (Liu et al., 2024) and a video LLaVA-Video (Zhang et al., 2024d). While we remember pretraining commonsense as noted in Table 6, we haven't explored improving other capabilities like math and science reasoning Yue et al. (2024). The scope of the paper, however, is to analyze what kinds of data improve spatial performance, and not a large-scale training of a new MLM. Additionally, further analysis on more MLMs (Dubey et al., 2024; Deitke et al., 2024) might be beneficial.

**Future work.** Although our study focuses on evaluating the spatial reasoning capabilities of MLMs, it can be extended in various avenues. For instance, to determine the kinds of embodied applications that benefit from improved complex spatial reasoning. We do see gains in accuracy in route planning in VSI-Bench (Yang et al., 2024a), which should help embodied navigation. As a more direct evaluation, we also checked the action prediction accuracy (from a choice of going left/right/forward) for a given frame on the SPOC Easy-ObjectNav benchmark (Ehsani et al., 2024). Our model (SAT Dynamic) scores an accuracy of 51% compared to 40% for a model trained only on basic spatial questions. This suggests that embodied navigation might benefit from improved dynamic spatial reasoning. We leave a more thorough evaluation in this direction as future work. Further, leveraging the interactive nature of our scenes with perfect 3D information could facilitate more explorations in dynamic and causal reasoning.

### A.2 Tuning details

We tune a widely used open-source MLM, LLaVA-1.5-13B Liu et al. (2024) for fine-tuning experiments. We LoRA-tune (Hu et al., 2021) with rank 256 and alpha 512. We full fine-tune the LLaVA-Video-7B model since it is smaller. We see minimal differences in trends by training with more lora parameters or with full fine-tuning as shown in Figure 7. To prevent catastrophic forgetting, we randomly sample examples from the LLaVA Instruct Tuning dataset Liu et al. (2024) with 40% probability while tuning with our synthetic QA pairs. We train all models to around 1 epoch of synthetic data. Due to memory constraints, we use a batch size of 8 (using gradient accumulation). We set a small learning rate of $5e^{-6}$ (due to our small batch size) with cosine annealing with 1K warm-up steps and a weight decay of 0. Training requires two 48GB NVIDIA GPUs, while inference is possible with one GPU. To demonstrate effect on a video benchmark, we tune another widely used video MLM, LLaVA-Video-7b (Zhang et al., 2024d). FOr LLaVA-Video-7b, the paper results are solely by training on SAT, without mixing any real orginal pretraining data. However, we also do compare the effect of mixing in real pretraining data as show it in Figure 4. We also show the sacling and

### A.3 Dataset Details

**Human Performance on our SAT** We conduct a human study with experts to measure the quality of our synthetic test data which is automatically generated. We observe that spatial awareness demands more mental power since one has to pay more attention and reason about how the orientation of the scene changed or would change based on an action. We conduct an expert human study, where we ask anonymized graduate students to answer 200 randomly sampled questions from our test set using the interface showed in Figure 10. We see that humans are 92.8% accurate on our SAT dataset. This is still a significant gap compared to the performance of best existing zero-shot MLM (around 53%). We note that zero-shot SAT accuracies on the synthetic test set are lower since synthetic image are often out of domain for MLMs. Hence, we most use SAT-synthetic to as diagnostic set to analyze performance gains after tuning the MLMs. We see tha gains on the synthetic test set are indicative of performance on the related task on real images. We will release the dataset on Huggingface. Our SAT real dataset was annotated by careful cross-checking by 4 graduate students. Our datasets will all be released publicly.

*More details on dataset creation*

We first take an apartment from ProcTHOR-10K and place the camera at a position where many objects are visible. We do this by randomly choosing 20 points to place the camera and then choosing the point with max objects visible.

**Normalizing the camera coordinates** In ProcTHOR Deitke et al. (2022), in the camera view, the $y$ coordinate is the height coordinate, which means the $y$ increases pointing upwards (*e.g.,* the ceiling has a greater $y$ than the floor). Hence, from the bird's eye view, the coordinates are $x$ and $z$. The rotation of the camera is such that it is always parallel to the $x$-$z$ plane. Hence the rotation is described as angle clockwise around the $y$-axis with the camera pointing to the positive z-axis as a 0-degree rotation.

Given a camera rotation, we normalize the view by translating to $(0,0)$ for $x$ and $z$ by subtracting the camera $x_0$ and $y_0$. Further, we rotate the $x$-$z$ plane such that the camera points to the positive $z$-axis.

For rotation, we use the formula:

$$R = \begin{bmatrix} \cos(a) & -\sin(a) \\ \sin(a) & \cos(a) \end{bmatrix}$$

Hence, the normalized $x'$, $z'$ for any object is computed using:

$$\begin{bmatrix} x' \\ z' \end{bmatrix} = R \cdot \begin{bmatrix} x - x_0 \\ z - y_0 \end{bmatrix}$$

The $y$ value remains unchanged since it is the height, which is not affected since we do not change the camera height.

Hence, finally, $x'$ goes negative to the left and positive to the right, $z'$ goes positive towards the depth and $y$ goes positive upwards from the floor level. We use the values of $x'$ and $z'$ to calculate relative relationships (left, right, in front of, and behind) as described below.

*SAT Static Spatial QAs*

**Relative spatial relations.** For instance, if the value of $x'$ for "chair" is lower than that of "table", the chair is to the left of the table. We can also compute the distance between objects. We randomly choose 3 objects. We compute the pairwise distances using their $(x, y, z)$ 3D coordinates. Based on whether object 1 is closer or further to object 2, we make QAs like "Is the couch closer to the lamp or the table?"

**Relative Depth.** Similarly, if the value of $z'$ for, say, "lamp" is greater than that of "couch", we say the "lamp" is further away from the camera than the couch.

*SAT Dynamic Spatial QAs*

**Egocentric Movement.** We first choose an image frame. Next, we first choose to rotate left or right from angles 20, 30, 40, 50, 60 chosen randomly. We use the `controller.step(action='RotateRight', degrees=angle)` function in the AI2THOR Kolve et al. (2017) platform. Next, we move forward with probability 0.5 by a random distance from 20 to 40 centimeters (`controller.step(action='MoveAhead', moveMagnitude=dist`). We capture the next frame from this final position. We also formulate camera movement questions. Note that camera movement and rotation are separate since a camera can be moving left while rotating right. For camera movement, we follow a translate the camera left/right in a random direction by a random distance. Specifically, we first rotate by an angle between 45 to 90 degrees to the left/right, then move forward by a random distance and then rotate right/left (right if previously rotated left in the first step and vice versa) by an angle between 90 and 135 degrees. Sometimes, we do not move or rotate.

**Object Movement.** We first choose an object visible in the scene with at least a certain bounding box area to make sure the object is not too small or non-salient. Next we decide to

move the object by a random distance sampled from 0.25 to 0.5 meters in a random direction if possible. We use the `PlaceObjectAtPoint` function in the AI2THOR Kolve et al. (2017) platform. Sometimes, we do not move any object. Sometimes the camera as well as an object moves. The agent needs to disentangle the two to answer questions correctly.

**Allocentric Perspective.** We choose a point on the 2D image that is not too close to the margsns (in between 0.2-0.8 normalized width and height of the image). We use `GetCoordinateFromRaycast` action in AI2Thor Kolve et al. (2017) to get the 3D location of the point. We try a few points since some points may not possible to navigate to and pick one possible point. We randomly turn left or right by 90 degrees. Next, we check whether an object is left/right based on the method described in Section A.3.

**Goal Aiming.** We compute the angle between a randomly chosen object with the camera assuming looking straight is 0 degrees. If the angle of an object is $-\alpha$, we say we have to "turn left by $\alpha$ degrees" and "right by $\alpha$ degrees" otherwise. When $|\alpha - 0| \leq \epsilon$, we say we have to look "roughly straight". We define $\epsilon = 10$. Note it is very hard for a human or machine to judge the exact degrees to turn from one single image, hence, we give multiple choices where the difference in choices are between left/right and hence, the machine really only has to decide between them and not the exact angles to turn. We use the following equation to calculate the angle of an object with the camera. First we normalize the object coordinates to the camera coordinates, $(x_0, y_0)$, and then calculate the angle, $\alpha$.

$$\begin{bmatrix} x' \\ z' \end{bmatrix} = R \cdot \begin{bmatrix} x - x_0 \\ z - y_0 \end{bmatrix}$$

$$\alpha = \arctan\left(\frac{x'}{z'}\right)$$

**Action Consequence.** Here, we just compute at the objects we got close to or further away while taking the action for the action sequence task or the perspective task. We formulate questions based on that. Note that most of the time we get close to objects in the scene and hence we random subsample such cases.

*Pseudo-annotated QAs*

We use DepthAnything (Yang et al., 2024b) to estimate depth of the scene and bounding box annotations in GQA (Hudson & Manning, 2019a) and VG (Krishna et al., 2017) to formulate static spatial QAs. To make spatial QAs, we first choose three objects in the scene based on which we make questions as defined in Section 3.1. We use the following heuristics to first choose the three objects to reduce noise:

– Too small objects: We do not choose objects where the bounding box area occupies less than 10% of the area. We observe this removes a lot of noisy annotations and very non-salient objects.

– Partially occluded objects: We do not choose objects if they are partially occluded by another object (based on the bounding box annotations).

Hence, we choose three clearly visible objects in the scene, estimate it's depth and and the center point location. Based on this 2.5D information, we formulate questions about their depth and relative locations (left/right, behind or in front of).

*Evaluation details*

**Evaluating on VSR and GQA-Sp**

For VSR (Liu et al., 2023), we have captions like "truck in front of airplane." We re-formulate them to into quetsion-answer pairs like: "Is truck in front of airplane? Answer yes or no" to keep all evaluation formats consistent. This ensures performance differences are due to the difficulties of spatial relationships in the datasets and not the question-answering format.

**Evaluating on VSI-Bench**

Table 8: Datasource for LLaVA-Video on static and dynamic

| | SAT Real | CVBench | | | | | BLINK | | | |
|---|---|---|---|---|---|---|---|---|---|---|
| | | Count | 2DRel | 3DDep | 3DDist | Avg | MV | RelDep | SpRel | Avg |
| LLaVA-Video | 53.5 | 59.3 | 77.0 | 71.3 | 54.7 | 65.2 | 39.1 | 55.6 | 75.5 | 56.7 |
| + SAT Static | 51.6 | 66.2 | 85.7 | 90.5 | 84.3 | 81.2 | 54.9 | 62.9 | 73.4 | 63.7 |
| + SAT Dynamic | 63.4 | 66.2 | 81.2 | 88.2 | 79.3 | 78.4 | 48.1 | 66.1 | 73.4 | 62.6 |

Table 9: Datasource for LLaVA-VIdeo on long video VSI-Bench. Row b and c are trained purely on simulations, while row d is simulation and orignal pre-training data (LLaVA-OV data) mixed in during training to prevent forgetting.

| | VSI-Bench (vid) | | | | |
|---|---|---|---|---|---|
| | Rel Dist | Rel Dir | Rt. Plan | App. Order | MC Avg |
| a. LLaVA-Video | 43.9 | 42.0 | 33.5 | 32.4 | 38.0 |
| b. + SAT Static | 43.5 | 43.4 | 34.5 | 30.3 | 37.9 |
| c. + SAT Static + Dyn | 47.7 | 42.0 | 35.6 | 36.1 | 40.4 |
| d. + SAT Dyn + LLaVA-OV | 47.9 | 39.6 | 38.7 | 40.6 | 41.7 |

For VSI-Bench (Yang et al., 2024a), we only use the MC split and not the numerical splits. We believe the numerical splits requiring estimation of exact distances and dimensions from RGB images alone without camera and depth information is difficult and out of scope for SAT since SAT focuses on relationships in images and dynamic scenes and not exact depth and distances.

**Question-answer format**

We frame each questions as a binary choice following existing benchmark standards (Fu et al., 2024b; Tong et al., 2024), and to prevent the MLM from choosing shortcuts by eliminating obvious wrong choices in multiple choices (Cai et al., 2024).

Following LLaVA Liu et al. (2024) convention, we use <image> tokens to represent images. This is the exact prefix we use.

```
A chat between a curious human and an artificial intelligence assistant.
The assistant gives helpful, detailed, and polite answers to the human's
questions.
```

Next, we add the question prompt to the prefix:

```
###Human: <im_start><image><im_end> Human: Answer in natural language.
Is the person facing the frisbee? Choose between the following options:
yes, or no.###Assistant:
```

For questions with two images, we simply have <im_start><image><image><im_end> in the image part of the prompt.

The prefix with "A chat between a ..." is something we found to be very important for LLaVA performance. Hence, we append this prefix to the question both, when tuning and testing. Further, we also found performance improvements when we specify the answer choices in text like "choose between 'left' or 'right'" than asking the model to choose an answer option letter (like A or B) or number (like option 1 or 2). We randomize the answer choice order during evaluation. Since SAT Real is small, we perform circular eval, where we prompt the model with both ordering of choices and average the performance. This reduces randomness and is robust. We also note a higher variance in performance between different training seeds on BLINK due to the small size of the dataset. However, the trends remain the same. We will release all checkpoints, the training script, and the best tips and tricks in the training schedule.

Table 10: Effect of dynamic data keeping the data size the same as static at different scales. This is to judge the effect of types of simulated spatial data. We see that adding dynamic maintains static spatial reasoning while improving dynamic and video spatial reasoning.

| | SAT Real | CVBench | | | BLINK | VSI-Bench |
|---|---|---|---|---|---|---|
| | | CVB 2D | CVB 3D | Avg | Avg | MC Avg |
| LLaVA-13B | 41.58 | 52.95 | 50.40 | 51.68 | 57.13 | - |
| +Stat 127K | 40.08 | 69.40 | 76.42 | 73.40 | 59.84 | - |
| +Stat+Dyn 127K | **51.26** | **73.37** | **77.33** | **75.99** | **63.26** | - |
| LlaVA-Video-7B | 53.45 | 67.30 | 63.00 | 65.15 | 56.73 | 37.95 |
| +Static 50K | 49.94 | **74.55** | **85.00** | **79.77** | 59.39 | 38.73 |
| +Stat + Dyn 50K | **59.28** | 74.13 | 82.08 | 78.11 | **62.34** | **40.36** |
| LlaVA-Video-7B | 53.45 | 67.30 | 63.00 | 65.15 | 56.73 | 37.95 |
| +Static 100K | 51.58 | **75.03** | **87.42** | **81.22** | **63.74** | 37.94 |
| +Stat + Dyn 100K | **59.13** | 73.50 | 85.25 | 79.38 | 61.09 | **38.63** |

Table 11: Most frequent correct and incorrect words

| Rank | Correct Words | Incorrect Words | Incorrect Words (excluding count) |
|---|---|---|---|
| 1 | stationery | 3 | 4x4 |
| 2 | pan | 5 | iced |
| 3 | books | 4 | wheeler |
| 4 | my | 4x4 | donuts |
| 5 | into | 12 | cookies |
| 6 | vase | 6 | tea |
| 7 | straight | iced | poles |
| 8 | magazine | 10 | office |
| 9 | signs | 11 | game |
| 10 | flowerpot | 8 | dogs |

### A.4 Extra ablations and analysis.

**Analyzing the semantic concepts where models fail after SAT tuning.** We wish to understand the semantic concepts where SAT tuning doesn't suffice. Hence, we we analyze the words that most often occur in the failure cases in contrast to the correct cases and show them in Table 11. Unsurprisingly, we see a high fraction of count words with higher numbers than 2 being mostly present. If we exclude the coutn words, we see a higher fraction of outdoor words. This is also unsurprising since our simulator only has indoor scenes. Using world models and more diverse simulators to improve our dataset is an exciting area of future work.

**Errors in relationship judgment compounded with lack of reasoning causes MLMs to perform worse than random on dynamic tasks** For certain splits such as perspective, or overall performance on SAT real, some accuracies are below random chance (random is 50%). In dynamic examples, the spatial position of an object often switches when a new perspective is taken—*e.g.*, a cup might be in the left side of the image, but switches to the right side after taking a new perspective. Weaker MLMs have bad spatial perception in general. However, even for MLMs with strong static relationship performance, they often answer what is currently visible (*i.e.,* cup is on the left) rather than what actually happens after the new perspective (*i.e.,* cup on the right). Examples are shown in Figure 5. For example, let's look at the performance of a model which has strong static spatial performance - Gemini-1.5-pro. If we look at the second row example, the surfer is currently on the left of the X marked point and hence the model answers left instead of reasoning about how it might switch when viewing from that perspective. Similarly, in row 3, and in row 7 where the door is currently visible on the left. While this analysis is from qualitative analysis, a more thorough analysis is left as future work.

**Adding in dynamic helps LLaVA-Video on dynamic and video spatial reasoning** As shown in Tables 8 and 9, adding in dynamic data helps LLaVA-Video on video spatial reasoning (Table 9) and on dynamic SAT Real in Table 8, while maintaining performance

on CVBench and BLINK. While scaling works better with mixing in dynamic as shown in Figure 4 in the main paper, we also analyze the effect of adding dynamic data while maintaining the same data size as static in Table 10. We see that for both models, adding in dynamic helps significantly on SAT Real. For LLaVA1.5 it helps overall. For LLaVA-Video, it helps on video and SAT real while maintaining performance on BLINK and CVBench.

**Quality of data is important. Hence, simulations are an effective source.** We wish to answer if the gains trivially come from just simply learning the formats. Hence, we run a noise baseline where we flip the answers 20% of the time. We see performance degrades significantly. We see a reduction of 10% on CVB and 18% on BLINK when training with just 20% noise. This means MLMs are not trivially learning the format of questions or object/attribute -name-based biases. Rather, the spatial answer quality is important.

| Image(s) | Question | GT Answer | LLaVA Answer | LLaVA + SAT Answer | Gemini1.5pro Answer |
|---|---|---|---|---|---|
| | were any of the objects in the initial frame that you can still see in the second frame moved from their original positions? | chair was moved left and away from the camera | ❌ | ✅ | ✅ |
| | For someone at the x marked point and facing 90 degrees to the left, will the person surfing be to their left or right? | right | ❌ | ✅ | ❌ |
| | If i stand where the woman is and face the same way as her, is the truck to her left or right? | right | ❌ | ✅ | ❌ |
| | if i climbed on top of the mountain on the left, would the people in the scene appear bigger or smaller? | smaller | ❌ | ✅ | ✅ |
| | if i stand in the shower facing the faucet, will i be able to see the towels without the mirror? | no | ✅ | ✅ | ✅ |
| | how should the bears at the back rotate to face the bear in front? Rotate left by 90 degrees or right by 90 degrees? | Rotate left by 90 degrees | ❌ | ✅ | ❌ |
| | if i am standing in the centre of the room, facing the camera. is the door to my left or right? | right | ❌ | ✅ | ❌ |
| | If I rotate right and move forward, will the window move closer to me? | yes | ❌ | ✅ | ✅ |

Figure 5: Some qualitative results of spatial question answering on some of our SAT Real Dynamic Benchmark

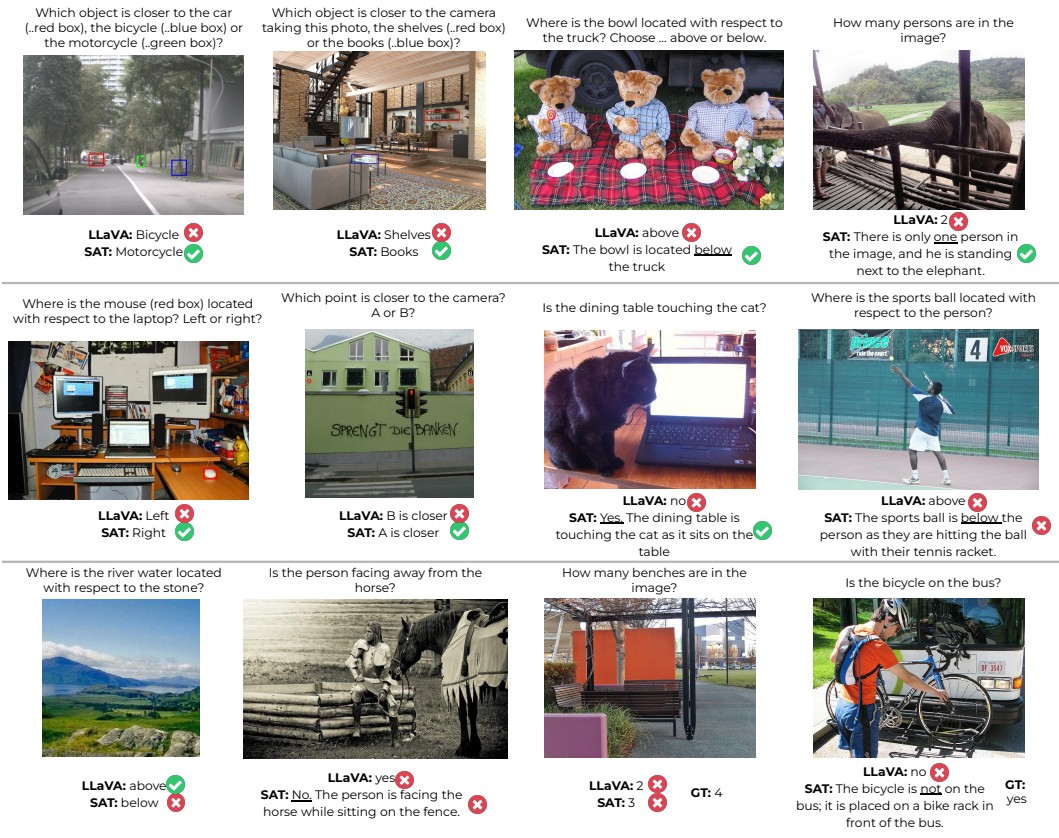

Figure 6: Some qualitative results of spatial question answering comparing baseline LLaVA on real benchmarks.

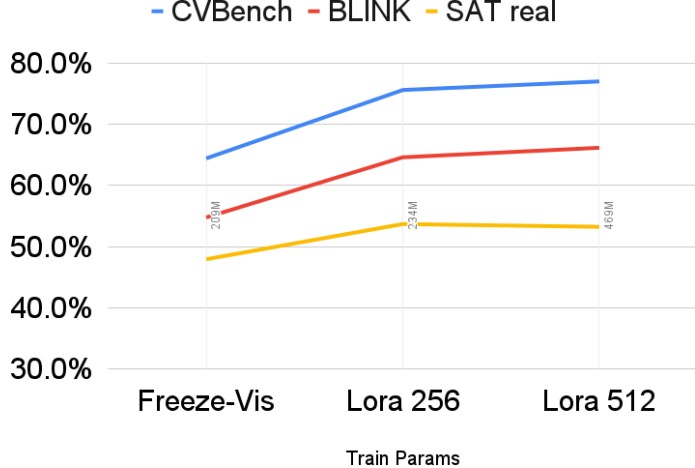

Figure 7: Effect of scaling up more tuning parameters. Tuning vision encoder is critical - showing visual features are a current bottleneck for spatial reasoning. However, tuning more LLM parameters yields incremental returns.

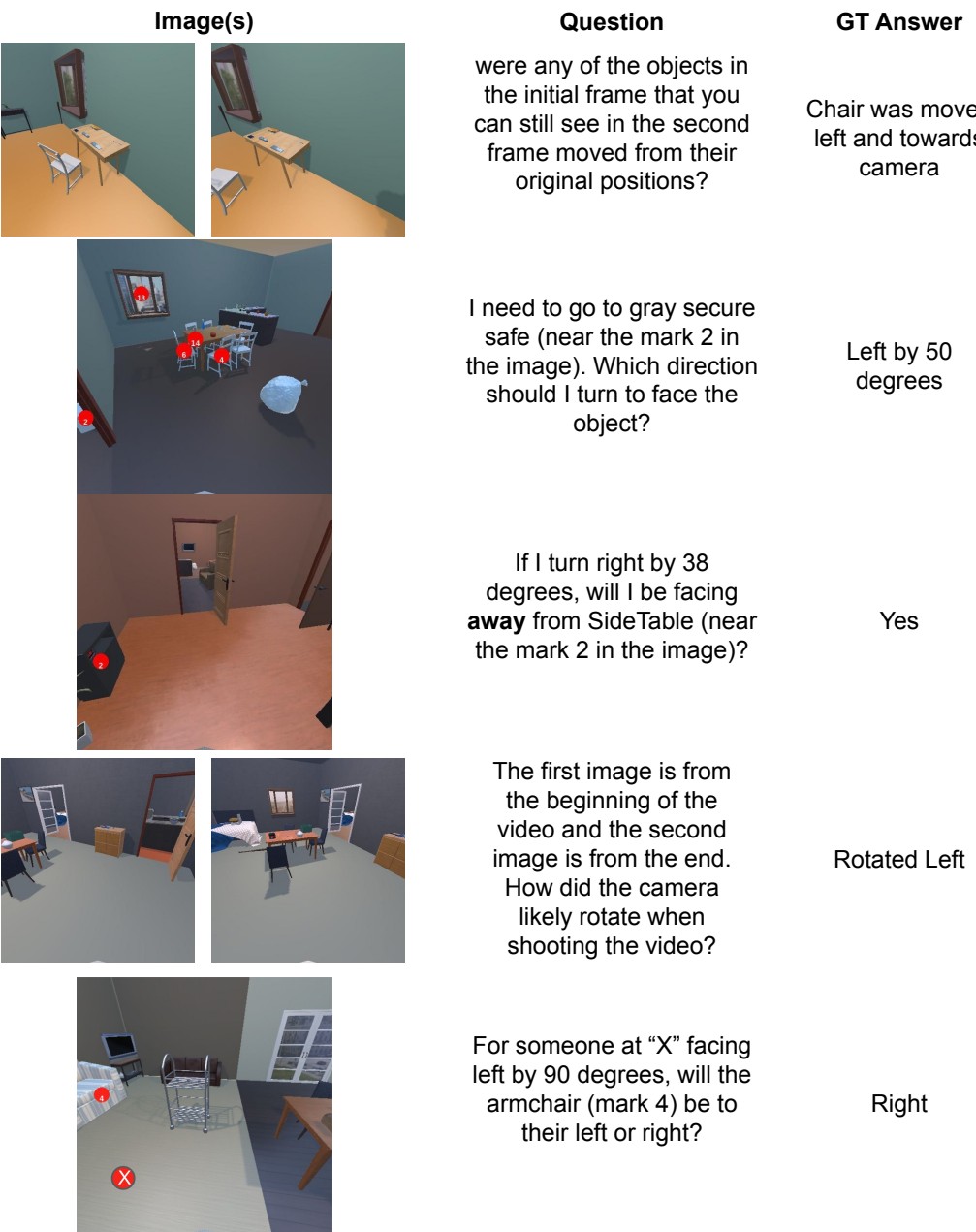

Figure 8: Qualitative examples of our synthetic SAT dynamic data

**Depth**
**Q:** Which object is closer to camera, countertop or garbage can?

**Spatial Relationship**
**Q:** Is wine bottle (mark A) to the left or right of dresser (mark C)?

**Answer:** garbage can

**Answer:** right

Figure 9: Qualitative examples of our synthetic SAT static data

**Instructions**

Thanks for participating in this HIT!

For the image(s) shown below, please answer the question to the best of your ability.

There may be one or two images depending on the question.

If the question seems ambiguous, please answer based on what you think is most likely meant.

Please **take your time** and do the HIT's diligently. We will be monitoring for rapid random clicks and will reject your HIT if we find evidence for rapid clicking.

## Answer the question to the best of your ability for the image(s) shown.
*Select the correct answer from the choices.*

First/Initial/Original Image

Second Image

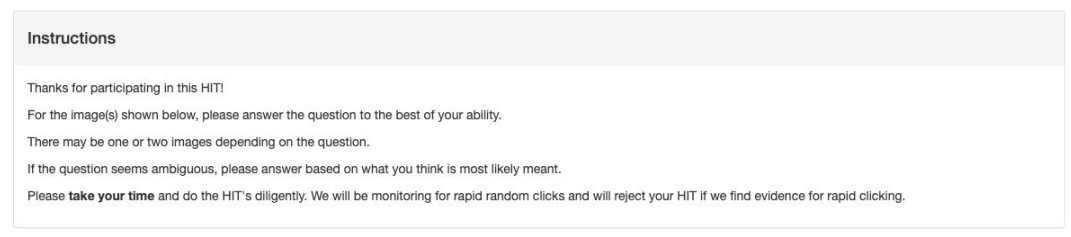

Question: The first image is from the beginning of the video and the second image is from the end. How did the camera likely move when shooting the video?

rotated left and moved forward ○
rotated right and rotated right ○

Figure 10: Interface to compute human accuracy for SAT.

