# OpenReview forum: "SAT: Dynamic Spatial Aptitude Training for Multimodal Language Models"
_colmweb.org/COLM/2025/Conference — COLM 2025_

### Official Review · Reviewer_VDnM · 2025-04-13

**Rating:** 7
**Confidence:** 3
**Ethics Flag:** 1

**Summary:**

This paper introduces new data for training and evaluating spatial understanding of MLMs.

The authors generate a large dataset with synthetic images, and collect a very small (150 QAs) test set with real images.

The goal is to improve and evaluate the abilities to understand static and dynamic (i.e. effect of motions) spatial properties.

**Questions To Authors:**

None.

**Reasons To Accept:**

Strengths:
--

(S1) Sound motivation. Fills a gap in the literature and space of available datasets.

(S2) Scalable approach. The data is generated procedurally without manual per-example explicit supervision.

(S3) The paper examines of the data, in particular which types of data improve different skills (static and dynamic understanding).

(S4) The results show benefits on real-image benchmarks (despite the new training data being synthetic).

(S5) Generated data validated by testing humand performance (Appendix A.3)

---

Reasons to accept:

- valuable contribution as detailed above

- I do not see any clear flaw to this work, although I might have missed something (I am not an expert on this specific topic).

**Reasons To Reject:**

Weakness/suggestion for improvementy:
--

(W1) What the authors generate are **QAs for instruction tuning**. This is only made clear in Section 3 but should be clarified earlier. I could imagine a similar pipeline to be used to generate structured annotations or full-text captions describing the same synthetic scenes.

---

Minor:
--

L148 "*MLMs struggle with counting*": the reference of (Goyal et al., 2017) doesn't seem appropriate/relevant, certainly not up to date.

L263, 270, 288, 323, 329, etc. (also in the Appendix): need a dot at the end of the paragraph headers.

L676: unresolved reference ("*Figure ??*")

L680 "*batch size of 8 (using gradient accumulation*": is this 8 before accumulation? then what is the effective batch size?

L695 "*we most use ... to as ...*" -> mostly ... as

L696 "*We see tha gains*" ???

L811: no dot at the end of the section title.

Please proof-read the whole appendix.

---

> ### Author Response · Authors · 2025-06-03
>
> Thank you for your positive and helpful review! We are glad you find this work valuable, scalable, and that the results show benefits on real-image benchmarks.
>
> Yes, this pipeline can also be used to generate structured annotations (that benefit more detailed reasoning) or captions. We completely agree and will clarify that in this work, we are focusing on QAs for instruction tuning.
>
> Thank you again for catching the typos. We will update the citations and the formatting issues.
> The batch size of 8 is after gradient accumulation—so, a batch of 1 with 8 gradient accumulation steps leading to an effective batch size of 8. We will clarify this.

---

> > ### Comment · Reviewer_VDnM · 2025-06-07
> > **OK**
> >
> > (Sending this dummy message because I keep getting daily reminders that there's a response with no reply.)

---

### Official Review · Reviewer_MPJk · 2025-05-12

**Rating:** 6
**Confidence:** 2
**Ethics Flag:** 1

**Summary:**

This paper introduces SAT (Spatial Aptitude Training), a simulated dataset for training multimodal language models (MLMs) on spatial reasoning tasks. The authors create 175K QA pairs across 20K scenes, including both static spatial relationships and dynamic spatial reasoning tasks that involve reasoning about how egocentric and object motions affect spatial relationships. They also create a small real-image test set and evaluate on existing spatial benchmarks. Their results show that training MLMs on simulated data improves spatial reasoning capabilities on real images.

**Questions To Authors:**

- How well would your approach generalize to real-world applications with more complex scenes than those in your simulator? Have you tested on more challenging real-world images? How much do you think the improvements from dynamic data would transfer to embodied AI applications like navigation in the real world?
- Your synthetic test accuracy is lower than real-world test accuracy for many models. Why do you think this is the case?
- Have you considered using more diverse simulators beyond ProcTHOR to increase the variety of your training data?

**Reasons To Accept:**

- Important research direction that addresses a real gap in MLMs. Spatial reasoning is crucial for many applications but current models struggle with it, especially with dynamic spatial relationships. The authors make a good case for why this matters based on cognitive science literature.
-  The results look impressive, with significant gains over baselines and sometimes even over proprietary models like GPT4-V and Gemini. Comprehensive evaluation across multiple datasets (both their own and existing ones) and model types. They test on four static spatial benchmarks, their own dynamic test set, and a video benchmark. The ablation studies reveal interesting insights.

**Reasons To Reject:**

- Limited discussion of failure cases or analysis of where their approach breaks down. While they mention that camera movements remain challenging (lines 266-269), a deeper error analysis would help understand the limitations of simulation-based training.
- Concerns about sim-to-real transfer and out-of-distribution scenarios. The paper doesn't adequately address how their simulated training data might not cover the diversity of real-world spatial scenarios. The test set is small (only 150 image-QAs), which makes me wonder about robustness.
- The paper presents better performance on their own dataset than existing baselines, which is somewhat expected. While they do show gains on external benchmarks as well, it would be helpful to see a more detailed analysis of the types of questions/scenarios where their approach fails on those benchmarks.

---

> ### Author Response · Authors · 2025-06-03
>
> Thank you for the positive and helpful review! We are glad you find this work crucial for many applications and that the results are impressive and reveal interesting insights.
>
> Here are some answers to address your concerns:
>
> **1. A deeper discussion of failure cases or where the approach breaks down beyond camera movements and discussions on out-of-distribution scenarios.**
>
> Thank you for this great suggestion! Some failure cases are shown in the appendix in Figure 6. First, we see that abstract relationships like “looking away” are hard to capture in simulation, and hence models struggle with them (Appendix Figure 6, bottom row, 2nd from the left). Please see reply to 2SN7 point 2 for more details.
>
> To provide a more quantitative analysis, we compute the top 10 question/scene words that are seen in incorrect answers vs correct answers and present them below.
>
> | **Rank** | **Correct Words** | **Incorrect Words** | **Incorrect Words (excluding count)** |
> |----|----|-----|---|
> | 1 | stationery        | 3      | 4x4  |
> | 2 | pan | 5       | iced  |
> | 3 | books    | 4     | wheeler |
> | 4 | my       | 4x4    | donuts   |
> | 5 | into   | 12    | cookies    |
> | 6 | vase  | 6     | tea      |
> | 7 | straight          | iced     | poles   |
> | 8 | magazine          | 10  | office    |
> | 9 | signs             | 11    | game    |
> | 10 | flowerpot         | 8        | dogs  |
>
> First, we see that models continue to struggle with counting, as denoted by the number words in the “incorrect words” column. This is a well-known limitation in the community, and further research is needed to improve counting. Since our focus is on spatial reasoning, if we exclude counting words to analyze the top spatial failure categories (third column), we see that models fail on smaller objects in the scene, like donuts and cookies. Extending our pipeline with diverse assets and scenes is an exciting future work avenue.
>
> We will add this discussion to the camera-ready paper.
>
>
> **2. Transfer to real-world spatial scenarios that are different from what the simulation can cover.**
>
> Great point. We do see that the spatial knowledge transfers to more complicated real-world scenes and scenarios, as demonstrated by results on VSI-Bench, which contain more complicated spatial scenarios in videos like “route planning” than the reasoning in SAT.
>
> Following your suggestion, as another challenging evaluation, we **evaluate our model on MME-RealWorld-Lite** since it contains outdoor environments, and dynamic human and vehicle interactions, charts and graphs. We see improvements there too, notably across some spatial categories shown below. Please see the reply to 2SN7 point 2 for more details.
>
> | Model              | Overall | Motion Avg (outdoor) | Interaction (outdoor) | Position | Reasoning Avg | Perception Avg |
> | ------------------ | :-----: | :------------------: | :-------------------: | :------: | :-----------: | :------------: |
> | LLaVA‑Vid‑7B       |   35 %  |  26.8 %  | 24.5 % |   34 %   | 36.8 %    |  35.0 %  |
> | LLaVA‑Vid‑7B + SAT |   39 %  |  33.2 %  |  31.5 % |   42 %   |     39.7 %    |     39.6 %     |
> | *Delta*    |  *+4 %*  |  *+6.4 %* |   *+7 %* |   *+8 %*  |    *+2.9 %*    |   *+4.6 %* |
>
> A great avenue for further study is on such complicated dynamic scenarios, since significant room for improvement remains. We will include these results and a discussion on this in the camera-ready
>
> **3. Transfer of improvements to embodied AI applications like navigation in the real world**.
>
> We tested on route-planning on VSI-Bench (see Table 4, column Rt. Plan), which deals with planning a route for navigating in real-world scenes. Our spatial training helps in that regard, showing promise that such training might benefit real-world navigation. As another preliminary analysis, we also saw an increase in the accuracy of next-action prediction given a scene in the SPOC ObjectNav benchmark (see Appendix A1 lines 666-667). However, we agree that a deeper analysis on real robots and complex scenarios, such as self-driving, is a great avenue for future work that uses our models as foundational planning models. We will highlight this discussion in the camera-ready.
>
> **4. Sometimes, having lower synthetic test set accuracies than the real test set.**
>
> We posit that this may be due to the domain gap between real images and simulated images. In a zero-shot setting, MLMs pretrained primarily on real images may perform worse on synthetic images. The synthetic test set is simply a _diagnostic test set_ to see where improvements may be difficult/easy—i.e., the delta gains after tuning on SAT-train are reflective of real-world improvements (lines 270-275 in main paper). Please see reply to reviewer 2SN7, point 1 for more details.
>
> This is an insightful point, and we will add this clarification to the camera-ready.

---

> > ### Comment · Reviewer_MPJk · 2025-06-08
> >
> > Thank you for your response. It mostly fixed my concerns, and I would like to keep my positive ratings.

---

### Official Review · Reviewer_2SN7 · 2025-05-12

**Rating:** 7
**Confidence:** 3
**Ethics Flag:** 1

**Summary:**

The paper investigates whether simulation data can help improve static and dynamic spatial reasoning ability in LLMs. A data generation approach named Spatial Aptitude Training (SAT) is proposed to generate data in simulation environments characterizing both static and dynamic spatial reasoning. Training on SAT data shows that it helps with multiple aspects of spatial reasoning and even helps long-video reasoning. Some ablation studies are carried out to reveal the effective part of the SAT data.

**Reasons To Accept:**

- Extending spatial reasoning into dynamic settings is an important yet underaddressed direction. The data generation pipeline is carefully designed to address both static and dynamic scenes. Sampling data from simulation environments also mitigates the data scarcity issue in real-world VQA datasets.
- Comprehensive experiments and insights: experimental results justify the effectiveness of SAT data and shows its improvement in multiple aspects. The ablation study also provides interesting insights such as the importance of dynamic scenes and hints of scaling with these data samples.

**Reasons To Reject:**

- Limited Real-World Data: While SAT shows promising results in synthetic environments, the real-world test set is relatively small (150 image-QAs), which may not fully capture the complexities of real-world spatial reasoning. While there is also a synthetic split SAT-Synthetic, evaluation results on these two splits in Table 2 show different ranks and significant gaps in metric values.

- Generalization to Diverse Environments: The paper could further explore how well models trained on SAT generalize to a wider range of real-world scenarios, including outdoor environments or dynamic human interactions. Besides, evaluation could be improved by comparing to a base model that is fine-tuned on a similar amount of spatial visual questions to reflect the uniqueness and superority of SAT over general VQA generation pipelines.

---

> ### Author Response · Authors · 2025-06-03
>
> Thank you for the positive and helpful review! We are glad you highlighted the paper's direction in extending spatial reasoning to dynamic settings, the well-designed data generation pipeline, and the experiments that justify the effectiveness of the data, along with the insights from ablation studies.
>
> Below, we answer your concerns:
>
> **1. The amount of real data and differences in evaluation values between synthetic and real**
>
> *All our main evaluations are on real image benchmarks with over 9000 image-QA examples* from SAT-Real (150 im-qa), BLINK (400), CVBench (2638), VSI-Bench (2900), VSR (3000). Thank you for the feedback and we will add this detail to the camera-ready paper.
>
> SAT-Real and SAT-Synthetic may have different zero-shot metric values since the visual domain in real and synthetic is different. The synthetic test set is simply a _diagnostic test set_ to see where improvements may be difficult for dynamic splits—i.e., the delta gains are reflective of real-world improvements. This is discussed in lines 270-275 in the main paper. For example, higher gain on SAT-Synthetic-EgoM is also reflected by a higher gain on BLINK-MV (related capability of detecting camera movements)—LLaVA-video has a higher gain than LLaVA-1.5. Same as GoalAim to CV-Bench-2Drel (since pointing to the goal requires reasoning about left/right relations). More gain on SAT-Synthetic avg also leads to more gain on SAT-Real (LLaVA vs LLaVA-Video).
>
> Thank you for raising this insightful point! We will add this extended clarification to the camera-ready.
>
> **2. Generalization to a wider range of real-world scenarios, including outdoor environments or dynamic human interactions**
>
> Great point, our evaluation sets—CVBench, BLINK, and VSR—contain outdoor scenes and human relationships. We see that simulation training transfers well to these datasets, and qualitative results suggest (as shown in Appendix Figure 6) that the spatial knowledge transfers well to outdoor scenes as well.
>
> Following your suggestion, we **evaluate our model on MME-RealWorld-Lite** as another challenging benchmark containing outdoor environments, and dynamic human and vehicle interactions, charts, and graphs. Overall, we see a **4% gain** in performance compared to the baseline LLaVA-Video-7B when trained on SAT. Notably, if we select some spatial categories of interest, we see that understanding **interactions improve by 7%**, **motion by 6.4%**, and **positions of objects by 8%**. We will add all these results to the camera-ready.
>
> | Model              | Overall | Motion Avg (outdoor) | Interaction (outdoor) | Position | Reasoning Avg | Perception Avg |
> | ------------------ | :-----: | :------------------: | :-------------------: | :------: | :-----------: | :------------: |
> | LLaVA‑Vid‑7B       |   35 %  |  26.8 %  | 24.5 % |   34 %   | 36.8 %    |  35.0 %  |
> | LLaVA‑Vid‑7B + SAT |   39 %  |  33.2 %  |  31.5 % |   42 %   |     39.7 %    |     39.6 %     |
> | *Delta*            |  *+4 %*  |  *+6.4 %* |   *+7 %* |   *+8 %*  |    *+2.9 %*    |   *+4.6 %* |
>
> Some failure cases can be seen in Figure 6 in appendix. Abstract relationships including humans like “looking away” are hard to capture in simulations and, hence, models struggle with them (Appendix Figure 6, bottom row, 2nd from the left). Similarly, natural language noise ambiguities are difficult. E.g., in Figure 6, bottom row right-most image, the GT is “yes” for is the “bicycle on the bus”. But, technically, it is not “on” but “in front of.”
>
> **3. Comparison to a base model fine-tuned on a similar amount of spatial visual questions to reflect the uniqueness and superority of SAT over general VQA generation pipelines**
>
> Yes, this is a great point. We have this in Table 6 and described in Section 4.2. In summary, we trained pseudo-annotated spatial questions on the GQA dataset following other works that use pseudo-annotation as a spatial VQA generation pipeline (row b). We also trained on manual spatial annotations in datasets like VSR and 2.5VRD (row c). We see that SAT training performs better overall (row d and e) (+10% on CVB and BLINK). Details are in lines 305 to 316. These general VQA generation pipelines mostly have static data since automatically inferring the effect of dynamic actions on images is difficult. However, SAT is simulated and hence we can easily predict how the image changes with actions/motions. Further, even static spatial relations in SAT are highly accurate because the data is simulated—with perfect GT annotations. That said, more sophisticated ways to pseudo-annotate spatial QAs by lifting scenes to 3D and leveraging videos might be a great avenue of future research. We will make this discussion clearer in the camera-ready.

---

### Official Review · Reviewer_Q2hV · 2025-05-13

**Rating:** 6
**Confidence:** 3
**Ethics Flag:** 1

**Summary:**

This paper proposes a Spatial Aptitude Training (SAT) dataset to train vision-language models (VLMs) that can handle static and dynamic questions about images and videos. Using the dataset, they fine-tuned the LLaVA-13B and LLaVA-Video-7B models and demonstrated that the fine-tuned models outperform the baselines and commercial VLMs (GPT-4o/v, Gemini1.5-flash, and Pro).

**Reasons To Accept:**

- The proposed dataset contains dynamic questions that are new to the community.
- The experiment results show significant improvement compared to the baselines and commercial VLMs.
- Good qualitative examples to show the performance of different fine-tuned models and baselines.

**Reasons To Reject:**

- Besides the creation of synthetic datasets, the papers contain limited novelty in terms of the algorithm.
- The dataset is created from CVBench, BLINK, and VSI-Bench with pseudo-annotated Q&As. The paper lacks a detailed description of the dataset and its comparison to other similar datasets.
- No model novelty presented

---

> ### Author Response · Authors · 2025-06-03
>
> Thank you for the positive and helpful review! We are glad you highlighted our dynamic questions as new to the community and found our results to be promising.
>
> Below, we answer your concerns:
>
> **1. Dataset created from CVBench, BLINK, and VSI-Bench with pseudo-annotated Q&As.**
>
> The training set doesn’t include any examples from CVBench, BLINK, or VSI-Bench. These are used for testing only. The training set is our synthetic SAT dataset. (Please see section 4, lines 224, 225), We will clarify this detail in the camera-ready.
>
> **2. A more detailed description of the dataset and its comparison to other similar datasets.**
>
> Thank you for the feedback, and we will make the comparisons more explicit in the paper.
>
> Dataset details are in Section 3. Briefly, the database is a Visual Question-Answering dataset. It contains both static (127K) and dynamic (48K) portions, for a total of 175K QAs. The dataset is generated synthetically as described in Figure 2 using 175K scenes composed of 1K object class assets in ProcTHOR. The details on the individual splits are in Section 3. Please let us know if any other detail is needed. We will also include a word cloud and object class histogram in the camera-ready.
>
> Comparison to other datasets is in Table 1 and Related Works lines 91-105. The closest related datasets are SpatialVLM (not publicly available) and SpatialRGPT, both of which cover static relations and not dynamic scenes with multi-step spatial-reasoning pseudo-annotated on real images. Our focus on dynamic scenes and perfect 3D information from simulation distinguishes SAT from these pseudo-annotated real-image static datasets. Compared to other 3D point-cloud-based datasets, we have 2D images or multiple images since most state-of-the-art MLMs are tuned to take image inputs. Images are also cheaper for our use case, since recomputing 3D scans for dynamic scenes with moving objects is expensive.
>
> We will clarify these details more explicitly in the camera-ready.
>
> **3. Model and algorithm novelty.**
>
> We saw that re-using the large-scale pretrained model without changing the architecture led to strong generalization across multiple benchmarks since it preserves the pre-trained general vision-language commonsense (Table 5, main paper).
> The scope of the paper and its novelty lie in the generation pipeline of the simulated dynamic spatial reasoning dataset. We present fine-tuning on the dataset as a strong baseline. We completely agree that research on better ways to use our rich interactions in simulation to improve beyond simple fine-tuning is a great avenue for future work.

---

### Decision · Program_Chairs · 2025-07-07

**Decision:**

Accept

**Comment:**

This paper introduces SAT, a dynamic spatial aptitude training dataset that requires reasoning about ego and object motion, extending beyond the simple static relationships found in existing datasets. All reviewers have expressed positive opinions about the paper. We suggest that the authors provide a more detailed description of the dataset and further explore its generalization to real-world applications involving more complex scenes. Given the context, we recommend a clear acceptance of this paper.